# Alternation in Peripheral B Cell Subpopulations Is a Potential Biomarker for Autoimmune Diseases—A Cross-Sectional Study

**DOI:** 10.3390/diagnostics15131710

**Published:** 2025-07-04

**Authors:** Shao-Wei Ku, Tzu-Hua Fu, Huey-Ling You, Yu-Jih Su, Wan-Ting Huang

**Affiliations:** 1Department of Laboratory Medicine, Kaohsiung Chang Gung Memorial Hospital, Kaohsiung 83301, Taiwan; e050124@cgmh.org.tw (S.-W.K.); niniaiwa@cgmh.org.tw (T.-H.F.); youhling@cgmh.org.tw (H.-L.Y.); 2Department of Pathology, Kaohsiung Chang Gung Memorial Hospital, Kaohsiung 83301, Taiwan; 3Department of Biological Sciences, National Sun Yat-sen University, Kaohsiung 80424, Taiwan; 4Department of Medical Laboratory Sciences and Biotechnology, Fooyin University, Kaohsiung 83102, Taiwan; 5Department of Internal Medicine, Division of Rheumatology, Allergy and Immunology, Kaohsiung Chang Gung Memorial Hospital, Kaohsiung 83301, Taiwan; 6Department of Internal Medicine, Division of Geriatrics, Kaohsiung Chang Gung Memorial Hospital, Kaohsiung 83301, Taiwan; 7School of Medicine, College of Medicine, Chang Gung University, Taoyuan 33382, Taiwan; 8Institute of Biopharmaceutical Sciences, National Sun Yat-sen University, Kaohsiung 80424, Taiwan; 9Center for Mitochondrial Research and Medicine, Kaohsiung Chang Gung Memorial Hospital, Kaohsiung 833, Taiwan

**Keywords:** B lymphocytes, B lymphocyte subsets, autoimmune diseases, flow cytometry, biomarker

## Abstract

**Background:** Although autoimmune diseases differ in their pathogenesis, B cells play a central role in many of them, and alterations in peripheral B cell subpopulations have been observed. Therefore, we aimed to explore the possibility of peripheral B cell subpopulations as a biomarker for autoimmune diseases based on their alternation. **Methods:** We prospectively collected blood samples from 54 patients with various autoimmune diseases and 65 healthy controls. The percentages of B cell subpopulations were evaluated using flow cytometry. A scoring system was developed and the largest Youden’s index was used to determine the optimal cutoff point. **Results:** The frequencies of double-negative B cells and antibody-secreting cells were significantly higher in patients than in controls (median: 2.9% vs. 1.5%, *p* < 0.001; median: 3.6% vs. 2.1%, *p* = 0.001, respectively). Among the patients, those with systemic lupus erythematosus showed the most impact on the alteration of peripheral B cell subpopulations, which was correlated with disease activity. Furthermore, the scoring system effectively distinguished patients from healthy controls. The area under the receiver operating characteristic curves was 0.752 (95% confidence interval: 0.664–0.840), and the optimal cutoff value of ≥10 points yielded a sensitivity and specificity of 70.4% and 70.8%, respectively. **Conclusions:** Peripheral B cell subpopulations in patients with autoimmune diseases are significantly different from those in healthy individuals and can vary between diseases. Therefore, alterations in B cell populations may be a potential biomarker for diagnosing and evaluating autoimmune diseases.

## 1. Introduction

B cells are pivotal components of the adaptive immune system and are crucial in human immunity. They are involved in the production of antibodies and contribute to various immune processes, including cytokine production and antigen presentation. Autoimmune diseases encompass a heterogeneous group of disorders with distinct pathophysiological mechanisms. For example, the pathogenesis of spondyloarthritis (SpA) is particularly driven by the HLA-B27 antigen, activating T cells and producing pro-inflammatory cytokines such as tumor necrosis factor (TNF) and interleukin-17 (IL-17) [1]. In rheumatoid arthritis (RA), both T cells and B cells contribute to joint damage through autoantibody production [2]. Systemic lupus erythematosus (SLE) is characterized by immune complex formation, leading to inflammation and tissue damage across various organs [3]. Despite these mechanistic differences, B cells play a central role in the pathogenesis of many autoimmune diseases [4]. Research has identified significant differences in peripheral B cell subpopulations between patients with autoimmune diseases and healthy individuals [4,5,6]. Aberrant B cells can be sources of autoantibodies, which are crucial biomarkers for diagnosing, classifying, and monitoring disease activity in many autoimmune conditions [4]. In addition, B cells contribute to the pathogenesis of autoimmune diseases through cytokine secretion and antigen presentation [7].

Peripheral B cell subpopulations have emerged as potential biomarkers for autoimmune diseases. In patients with RA, double-negative B cells and CD95+ activated memory B cells have been shown to predict responses to B cell depletion therapy [8,9]. Furthermore, in patients with SLE, memory B cells and plasmablasts have been associated with early relapse following treatment with rituximab [10]. Similarly, in patients with rituximab-treated myasthenia gravis, memory B cells have been associated with clinical relapse [11]. Additionally, the proportion of plasmablasts within the B cell population has been suggested as a diagnostic marker for RA [12], and the count of immunoglobulin G4+ plasmablasts has been identified as a diagnostic tool for autoimmune pancreatitis [13].

However, despite substantial evidence demonstrating alterations in B cell subpopulations in patients with autoimmune diseases, the sensitivity and specificity of peripheral B cell subpopulation testing for evaluating these conditions remain relatively limited. Therefore, in this study, we aim to investigate the potential of B cell subpopulations as biomarkers for autoimmune diseases by examining the changes in these subpopulations. Additionally, we seek to develop a scoring system based on flow cytometry that could enhance the diagnosis of autoimmune diseases.

## 2. Materials and Methods

### 2.1. Study Patients

We enrolled 54 patients with autoimmune diseases, including 16 cases of SpA (29.6%), 14 cases of SLE (25.9%), 14 cases of RA (25.9%), 6 cases of axial spondyloarthritis (AS) (11.1%), 2 cases of sicca syndrome (3.7%) cases, and 1 case of systemic sclerosis (1.9%) and polymyositis (1.9%). The patients were followed up at the Rheumatology Outpatient Clinic of the Chang Gung Memorial Hospital in Kaohsiung, Taiwan, between January 2020 and December 2020. Patients with AS and SpA were diagnosed based on the modified New York criteria and the Assessment of Spondyloarthritis International Society classification criteria [14,15], respectively, whereas the others met the American College of Rheumatology/European Alliance of Associations for Rheumatology (ACR/EULAR) classification criteria [16,17,18,19,20]. All patients attended regular follow-ups and had stable disease for more than three months before B cell subpopulation analysis. We collected baseline data, including age, sex, treatment, and complete blood counts. Patients with recent infections, malignancy, pregnancy, or human immunodeficiency virus (HIV) infection were excluded from the study group. The control group consisted of 65 healthy individuals aged 20–65 who had not taken any medications for illnesses in the past 3 months. Individuals with pregnancy, allergies, autoimmune diseases, chronic liver diseases, chronic kidney disease, or HIV infection were excluded during the selection of the control group. The Chang Gung medical foundation institutional review board approved the study protocol (IRB201901509B0C501), and all participants provided informed consent.

We also retrospectively reviewed the medical records of 14 patients with SLE to obtain information on disease duration, the SLE Disease Activity Index 2000 (SLEDAI-2K) [21], and laboratory data within three months prior to the B cell subpopulation analysis. This included renal and immunological parameters. Creatinine levels and estimated glomerular filtration rate (eGFR, calculated using the Modification of Diet in Renal Disease formula) [22], as well as the spot urine protein/creatinine ratio (UPCR), were measured using standard laboratory techniques. Anti-double-stranded DNA (anti-dsDNA) antibodies were assayed using an enzyme-linked immunosorbent assay (QUANTA Lite, INOVA Diagnostics, San Diego, CA, USA) or fluorescence enzyme immunoassay (Phadia 250, Thermo Fisher Scientific, Waltham, MA, USA). Serum complement levels (C3 and C4) were measured using nephelometry (BN ProSpec System, Siemens, Munich, Germany).

### 2.2. Flow Cytometry Analysis

Whole blood was collected through venipuncture of the patient’s forearm vein, and samples treated with ethylenediaminetetraacetic acid were subjected to an eight-color flow cytometric immunophenotypic analysis, as in a previous publication [23]. A staining volume of 100 μL (2–3 × 10^6^ cells) of peripheral blood mononuclear cells separated through centrifugation was incubated with an antibody cocktail. Eight-color flow cytometry analysis was performed by staining the cells with the following antibodies: allophycocyanin (APC)-H7-labelled anti-CD45 (clone 2D1); V500-labeled anti-CD19 (clone HIB19); phycoerythrin (PE)-Cy7-labeled anti-CD38 (clone HIB7); V450-labeled anti-CD138 (clone MI15); APC-labeled anti-immunoglobulin M [IgM] (clone G20-127); fluorescein isothiocyanate (FITC)-labeled anti-immunoglobulin D [IgD] (clone IA6-2); PE-labeled anti-CD21 (clone B-ly4); and peridinin chlorophyll protein (PerCP)-Cy5.5-labeled anti-CD27 (clone M-T271). To accurately separate the fluorescent signals in multicolor analysis, we prepared “single-staining control samples” using the same fluorescently labeled antibodies to correct spectral overlap between different fluorescent dyes. We evaluated the specificity of the antibodies using known cell populations in blood. As gating strategies in Figure 1, mature B cells were classified into four subtypes based on the expression of CD21 and CD27 surface markers: CD21+ CD27− naïve B cells, CD21+ CD27+ memory B cells, CD21− CD27− B cells, and CD21− CD27+ antibody-secreting cells (ASCs). IgM and IgD were utilized to identify non-switched and switched memory B cells. The double-negative (DN) subsets were defined as CD21− CD27− IgD− B cells. CD38 and CD138 were employed to identify plasmablasts and plasma cells within the ASCs. B cells (CD19+) were classified into naïve (CD19+ CD21+ CD27−) B cells, including resting (CD21+ CD27− IgM+ IgD+) and IgM-negative (CD21+ CD27− IgM− IgD+) naïve B cells; memory (CD19+ CD21+ CD27+) B cells, including non-switched (CD21+ CD27+ IgM+ IgD+) and switched (CD21+ CD27+ IgM− IgD−) memory B cells; DN (CD19+ CD21− CD27− IgM− IgD−) B cells, including CD38− and CD38+ DN (CD21− CD27− IgM− IgD−) B cells, and ASCs (CD19+ CD21− CD27+), including early plasmablasts (EPBs) (CD21− CD27+ CD38 low/+), plasmablasts (CD21− CD27+ CD38+/++), and plasma cells (CD21− CD27+ CD38++ CD138+). Peripheral B cell subpopulations were identified and analyzed using a FACS Canto II flow cytometer (BD Biosciences, Heidelberg, Germany) equipped with three lasers (405 nm violet, 488 nm blue, and 647 nm red lasers) and InfinicytTM 2.0 software. The flow cytometry assay was validated according to the CLSI document H62 [24].

### 2.3. Statistical Analysis

The analyses were performed using SPSS software (v.22.0; SPSS Inc., Chicago, IL, USA). The normality of continuous variables was assessed using the Shapiro–Wilk test. As most variables were not normally distributed, we used the Mann–Whitney U test and the Kruskal–Wallis test to examine differences between two groups and four groups, respectively. Dunn’s post hoc tests were performed for multiple comparisons of the four groups. Pearson’s chi-squared test was performed for sex comparisons between two groups. Pearson’s correlation coefficient was used to explore the relationship between peripheral B cell subpopulations and disease activity parameters in the SLE group.

Furthermore, to assess the impact of B cell alterations on predicting autoimmune diseases, we developed a scoring system based on the quartile data of 16 variables (the variables in Table 1 except sex and age) from 65 healthy individuals. By dividing the data from the control group into four equal parts, we established key thresholds for each variable to differentiate the core data from outliers or extreme values. Points were then assigned based on the relative position of the data as follows: between the lower quartile (Q1) and the upper quartile (Q3) = 0 points, between Q1 to the lower fence (Q1 minus 1.5 interquartile range [IQR]) or Q3 to the upper fence (Q3 + 1.5 IQR) = 1 point, and below Q1 minus 1.5 IQR or above Q3 + 1.5 IQR = 2 points. A total of 16 variables were scored, with the total score ranging from 0 to 32. A higher point total indicated a more significant impact on B cell alterations. The area under the curve (AUC) was used to describe the ability of the model to differentiate patients with autoimmune diseases from healthy individuals. Youden’s index was used to determine the optimal cutoff point, and statistical significance was set at two-sided *p* < 0.05.

## 3. Results

### 3.1. Alterations of Peripheral B Cell Subpopulations in Patients with Autoimmune Diseases

The average age of the 54 patients and 65 healthy individuals was 45.9 years (range: 18.9–72.0 years) and 45.7 years (range: 25.7–69.9 years), respectively. There were no differences in age and sex distributions or the lymphocyte and B cell events obtained using flow cytometry. Regarding treatment in the patient group, nine patients with RA and all fourteen patients with SLE were on low-dose oral corticosteroids (equivalent dose of less than 0.3 mg/kg/day of prednisolone), while five RA patients were treated with methotrexate. Patients with SpA or AS primarily received non-steroidal anti-inflammatory drugs (NSAIDs). Three of the sixteen SpA patients were also treated with secukinumab, and four received sulfasalazine. Among the six AS patients, four were treated with sulfasalazine as well. No significant differences were observed in the control or patient groups when comparing males and females (all *p* > 0.05; Appendix A). The comparison of the base characteristics and B cell subpopulations based on flow cytometry data between the patients and controls is summarized in Table 1. The patients had lymphocyte percentages lower than the healthy controls (median: 27.2% vs. 30.8%, *p* < 0.001). The proportions of DN and ASC subpopulations were also different. The frequency of DN B cells was significantly higher in patients than in controls (median: 2.9% vs. 1.5%, *p* < 0.001, Figure 2a), including CD38− (median: 1.5% vs. 0.9%, *p* = 0.002) and CD38+ (median: 0.5% vs. 0.3%, *p* = 0.003) DN B cells. The analysis also revealed a higher frequency of ASCs (median: 3.6% vs. 2.1%, *p* = 0.001, Figure 2b), with a skewed distribution toward the EPB phenotype (median: 2.6% vs. 1.3%, *p* < 0.001) in patients compared with healthy controls. There was no significant difference in the frequency of memory B cells; however, the frequency of non-switched memory B cells in patients with autoimmune diseases was lower than in healthy controls (median: 4.8% vs. 7.4%, *p* = 0.001).

### 3.2. Comparison of Peripheral B Cell Subpopulations Among Patients with SLE, RA, and SpA

Furthermore, to investigate the differences in B cell subpopulations among patients with autoimmune diseases, we compared B cell subpopulations in three diseases with a higher number of patients, including SLE, RA, and SpA (Table 2). The percentages of DN B cells and ASCs were significantly different. Patients with SLE had a higher percentage of DN B cells than controls (median: 5.7% vs. 1.5%, *p* < 0.001) and those with SPA (median: 5.7% vs. 2.0%, *p* = 0.034) (Figure 3a). Further analysis of the DN B cell subsets showed that the percentage of CD38− DN B cells was significantly higher in patients with SLE and RA than in the control group (median: 2.4% vs. 0.9%, *p* = 0.004; median: 2.0% vs. 0.9%, *p* = 0.020, respectively). However, the percentage of CD38+ DN B cells was higher in patients with SLE than in the control (median: 1.3% vs. 0.3%, *p* < 0.001), RA (median: 1.3% vs. 0.4%, *p* = 0.001), and SPA groups (median: 1.3% vs. 0.5%, *p* = 0.047). Furthermore, the percentage of ASCs was statistically higher in patients with SLE than in controls (median: 7.2% vs. 2.1%, *p* < 0.001) and in those with RA (median: 7.2% vs. 2.1%, *p* = 0.005) (Figure 3b). A comparable pattern was observed for EPBs, with significantly higher percentages in patients with SLE relative to controls (median: 5.3% vs. 1.3%, *p* < 0.001) and RA patients (median: 5.3% vs. 1.6%, *p* = 0.004). In addition, the percentage of EPBs in the SPA group was higher than that in the controls (median: 2.5% vs. 1.3%, *p* = 0.005). However, no significant differences in plasmablasts and plasma cells were observed among the four groups. Furthermore, with increased percentages of DN B cells and ABS, patients with SLE had lower IgM-negative naïve and non-switched memory B cells when compared with the other groups (Table 2). However, despite the limited number of patients in each subgroup, the lupus subgroup reached statistical significance and power (the power for CD38− DN was only 0.78, but all others were >0.8) in its B cell analysis compared with the control group.

### 3.3. Association of Peripheral B Cell Subpopulations with Disease Activity Parameters in Patients with SLE

We further analyzed the relationship between disease activity parameters and B cell subpopulations in patients with SLE (Table 3). The anti-dsDNA titers of four SLE patients were measured using fluorescence enzyme immunoassay; thus, these data were excluded from the analysis. Their SLEDAI-2K ranged from 4 to 20, with a median score of 7. Notably, no significant correlation with total DN B cells was observed; however, the SLEDAI-2K was positively correlated with percentages of CD38+ DN B cells (r = 0.744; *p* = 0.002). Furthermore, the SLEDAI-2K had significant positive correlations with ASCs (r = 0.794; *p* = 0.001) and ASC subsets, including EPBs (r = 0.744; *p* = 0.002) and plasma cells (r = 0.760; *p* = 0.002). No significant association was observed between the SLEDAI-2K and white blood cell counts (r = 0.384; *p* = 0.176), lymphocyte frequency (r = −0.175; *p* = 0.550), or the frequencies of naïve and memory B cells (Table 3). In addition to SLEDAI-2K, anti-dsDNA titers were negatively correlated with IgM-negative naïve B cells (r = −0.672; *p* = 0.033), while the spot UPCR was positively correlated with plasmablasts (r = 0.873, *p* = 0.010) (Table 3).

Among the fourteen SLE patients, six were diagnosed with lupus nephritis based on a renal biopsy, while one patient with proteinuria declined the biopsy and was excluded from this analysis. The remaining seven patients exhibited no proteinuria. The Mann–Whitney U test revealed no significant differences in B cell subsets between SLE patients with and without lupus nephritis (all *p* > 0.05; Appendix A). However, due to the small sample size, these findings should be interpreted with caution.

### 3.4. A Scoring System for Peripheral B Cell Subpopulations

Furthermore, to evaluate if the alterations of the B cell subpopulations could classify individuals into the disease and control groups, we established a scoring system and determined the optimal cutoff value. The sum of points was calculated based on the scale of the percentages of B cell subpopulations exceeding the upper or lower limits. Patients with autoimmune diseases had a median score of 12 (range: 4–24) compared with the health controls, who had a median score of 8 (range: 2–21). The AUC of the score was 0.752 (95% confidence interval [CI]: 0.664–0.840; *p* < 0.001) (Figure 4), which was >0.75 and suggested satisfactory discriminative power. We estimated the threshold of scoring points to distinguish the disease from control groups with the Youden index. The cutoff value of ≥10 points yielded a sensitivity of 70.4%, a specificity of 70.8%, positive predictive values of 66.7%, and negative predictive values of 74.2%, with the largest Youden index of 0.412 and an odds ratio [OR] of 5.75 (95% CI: 2.61–12.69). The test outcomes for SLE, RA, and SPA were determined, which showed low positive and high negative predictive values (Table 4). Compared with the controls, patients with SLE had the strongest association with alterations of B cell subpopulations (OR: 31.5, 95% CI: 3.84–257.79).

## 4. Discussion

The present study demonstrated significant differences in the distribution of B cell subpopulations between the control and patient groups and distinct alterations among autoimmune diseases. The frequencies of DN B cells and ASCs were high, with the latter skewing differentiation toward EPBs in the patient group. Among the autoimmune diseases, patients with SLE revealed the most evident alteration of B cell subpopulations in the peripheral blood. Notably, some were correlated with disease activity, including CD38+ DN B cells, ASCs, EPBs, and plasma cells. Furthermore, we developed a scoring system based on the deviation of peripheral B cell subpopulations in patients with autoimmune diseases. A higher score was indicative of a greater difference from the healthy group. This scoring system helps to distinguish patients with B cell differentiation disorders, particularly those with SLE.

DN B cells are a subpopulation of B cells that do not express IgD and CD27 surface markers. They have been found in increased frequencies in patients with various autoimmune diseases [5,8,25]. Jenks et al. divided DN B cells into DN1 and DN2 subsets based on their expression of the markers CXCR5 and CD21 [26]. These subsets exhibit different functional properties and roles in autoimmune diseases. DN1 B cells expressing CXCR5 and CD21 may be a precursor of switched memory B cells [26]. However, DN2 B cells (CXCR5− CD21−) are more activated and pathogenic than DN1 B cells, with a higher propensity to produce autoantibodies and increase disease activity [26]. ASCs are another B cell subpopulation involved in the pathogenesis of autoimmune diseases [4,5]. This subpopulation contributes to autoantibody secretion, chronic inflammation, and tissue damage [4,27]. EPBs, instead of plasma cells, were the predominant subset detected in the present study. It may indicate that ASCs are involved in proliferation rather than antibody production in patients with autoimmune diseases undergoing treatment.

Our study, consistent with previous studies [5], showed the association between ASCs and SLEDAI-2K in patients with SLE. Notably, the subsets of EPBs and plasma cells play a pathogenic role [4,28]. Furthermore, CD38+ DN cells were positively correlated with SLEDAI-2K. Except for the lack of CD27 expression, the immunophenotype of CD38+ DN cells is similar to that of EPBs, which might indicate that CD38+ DN B cells represent an intermediate stage with skewed differentiation toward ASCs. Therefore, CD38+ DN B cells, like DN2 B cells, may have a propensity to cause disease activity in patients with SLE [26]. Our study also showed a positive correlation between the spot UPCR and plasmablasts. While direct evidence linking plasmablast levels and proteinuria is limited, our findings align with previous studies suggesting that both proteinuria and plasmablasts may be indicators of lupus activity [29].

Conversely, the frequency of IgM-negative naive B cells was statistically lower in patients with SLE and was inversely associated with anti-dsDNA titers. Downregulation of sIgM may occur while differentiating directly into ASCs [30,31]. This indicates that the low frequency of IgM-negative naive B cells in patients with SLE may be a dynamic change in B cell differentiation and contribute to the production of pathogenic autoantibodies and disease progression. A reduction in non-switched memory B cells is also observed in patients with SLE, with median levels comparable to those reported in previous studies [32]. This may be due to a higher proportion of autoreactive cells in this subset and their activation, migration into lymphoid tissues, and transition to class-switched memory B cells [33,34]. However, non-switched memory B cells were not significantly associated with any of the disease activity parameters, consistent with a previous study [29].

This study has several limitations. First, the small sample size and the limited number of cases in each disease subgroup may affect the representativeness of the findings. Second, the cross-sectional design precludes the assessment of longitudinal changes in B cell subpopulations and their responses to treatment. Third, as all patients were undergoing treatment, the results may not fully reflect the natural disease state. However, there is no direct evidence that methotrexate, sulfasalazine, or low-dose oral corticosteroids significantly affect the composition of B cell subpopulations. Although secukinumab may normalize B cell subpopulations [35], it was used in only a small proportion of SpA patients (three out of sixteen) in our study. Therefore, we believe that the impact of treatment on the distribution of B cell subpopulations is minimal. Fourth, sex hormones and X chromosome dosage are important modulators of humoral immunity [36]. These factors might also have an impact on B cells, even though our study did not observe any significant sex-related differences in either the control or patient groups. The fifth limitation concerns the methodology used in the present study. Qualifying the properties of B cell subsets and defining autoreactive B cells were difficult. Therefore, naïve B cells that do not express IgM, which represents the intermediate stage before full differentiation or dysregulated immune responses, could not be distinguished. Consequently, flow cytometric cell sorting to purify targeted cell subpopulations for further functional study should be addressed in future research. In addition, the scoring system of peripheral B cell subpopulations is an exploratory model, and further studies with variable selection and validation are needed to establish its reliability for assessing autoimmune diseases.

In conclusion, B cell subpopulations are significantly different in healthy and patient groups, making them a potential biomarker for diagnosing and evaluating autoimmune diseases. The alternation of B cell subpopulations in patients with SLE was the most significant in the disease group. We further developed a simple exploratory scoring system that can effectively discriminate between diseased and healthy individuals. Therefore, a further study including a larger number of patients with diverse immunological disorders and different disease stages is necessary to develop a valid and credible model.

## Figures and Tables

**Figure 1 diagnostics-15-01710-f001:**
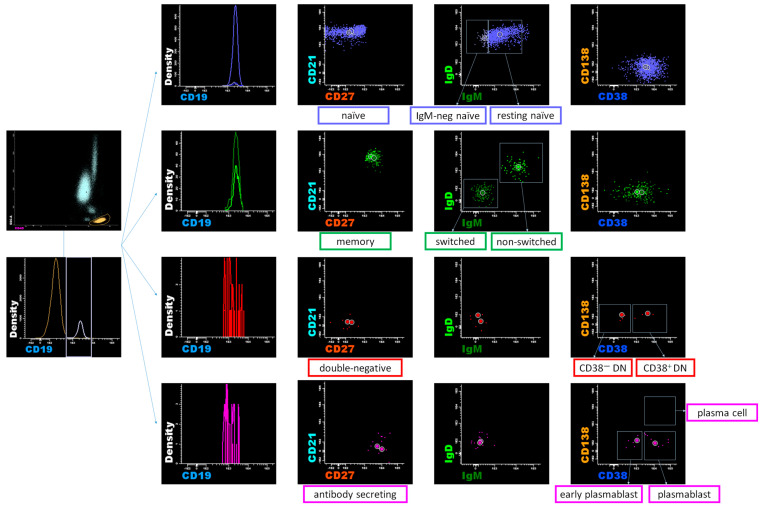
Gating strategies of B cell subpopulations.

**Figure 2 diagnostics-15-01710-f002:**
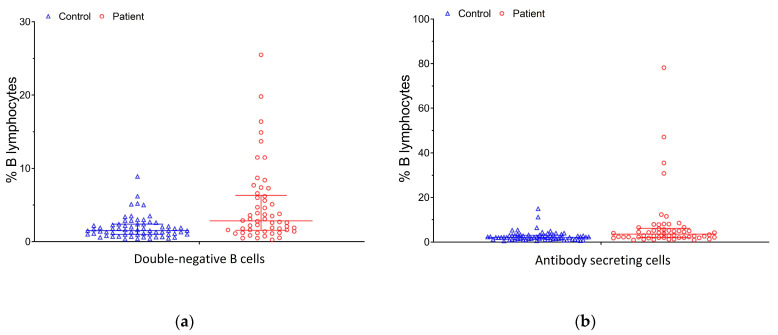
(**a**) DN B cell distribution of the control and patient groups; (**b**) ASC distribution of the control and patient groups.

**Figure 3 diagnostics-15-01710-f003:**
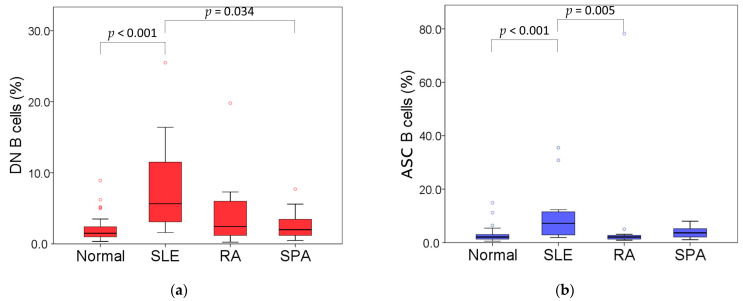
(**a**) Frequencies of DN B cells in total B cells in controls and three groups of patients; (**b**) frequencies of ASCs in total B cells in controls and three groups of patients.

**Figure 4 diagnostics-15-01710-f004:**
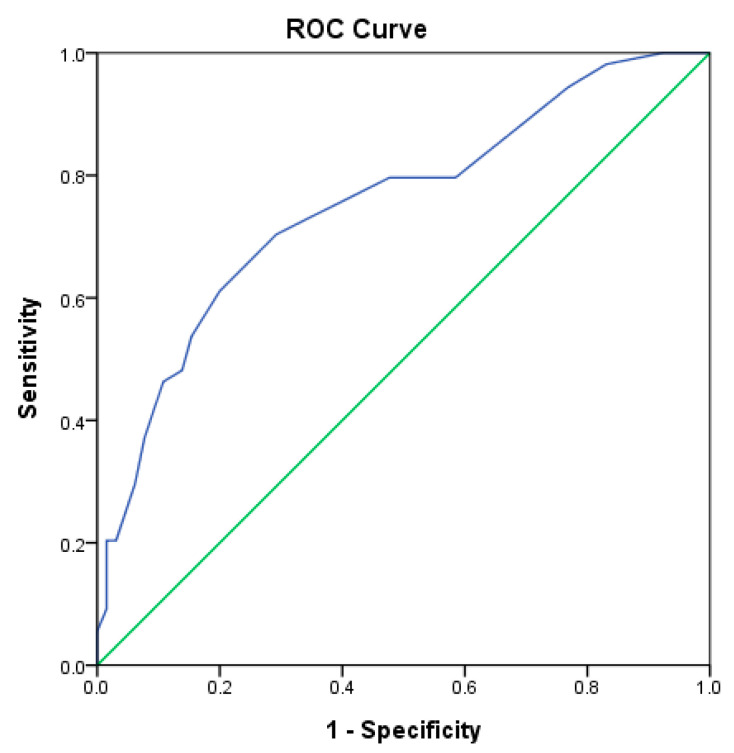
Receiver operating characteristic curve of the scoring system of peripheral B cell subpopulations, distinguishing the patients from the healthy controls. The blue line represents the ROC curve of the scoring system, while the green line indicates the performance of a random classifier.

**Table 1 diagnostics-15-01710-t001:** Baseline characteristics and peripheral B cell subpopulations of all patients and healthy controls.

	Patient (*n* = 54)	Control (*n* = 65)	*p*-Value
Sex (Male/Female)	25/29	28/37	0.725
Age	48.0 (16.7)	45.48 (22.7)	0.936
WBCs (1000/μL)	6.6 (2.8)	6.5 (2.3)	0.540
Lymphocytes (% WBC)	27.2 (14.1)	30.8 (8.2)	<0.001 *
B cells (% lymphocytes)	7.8 (7.0)	7.3 (5.0)	0.643
Naive (% B cells)	66.8 (26.1)	74.0 (20.3)	0.364
Naive Resting (% B cells)	38.8 (15.7)	42.6 (13.3)	0.356
IgM-negative Naïve (% B cells)	20.2 (22.0)	23.4 (20.8)	0.201
Memory (% B cells)	23.1 (21.7)	20.3 (17.7)	0.229
Non-switched (% B cells)	4.8 (8.9)	7.4 (8.3)	0.001 *
Switched (% B cells)	10.6 (12.7)	9.7 (8.5)	0.799
DN (% B cells)	2.9 (4.6)	1.5 (1.4)	<0.001 *
CD38− DN (% B cells)	1.5 (2.5)	0.9 (1.0)	0.002 *
CD38+ DN (% B cells)	0.5 (0.9)	0.3 (0.5)	0.003 *
ASC (% B cells)	3.6 (3.9)	2.1 (1.8)	0.001 *
EPBs (% B cells)	2.6 (3.1)	1.3 (1.3)	<0.001 *
PBs (% B cells)	0.4 (0.6)	0.3 (0.3)	0.692
Plasma cells (% B cells)	0.1 (0.8)	0.1 (0.1)	0.544

The continuous variables were analyzed using a Mann–Whitney U test and presented as median values (interquartile range). Sex was compared using Pearson’s chi-square test. * *p* < 0.05. Abbreviations: WBCs, white blood cells; DN, double-negative B cells; ASCs, antibody-secreting cells; EPBs, early plasmablasts; PBs, plasmablasts.

**Table 2 diagnostics-15-01710-t002:** The difference in the frequencies of peripheral B cell subpopulations (in total B cells) in controls and three groups of patients.

	Controls *n* = 65	SLE *n* = 14	RA *n* = 14	SpA *n* = 16	*p*-Value
Naive (% B cells)	74.0 (19.6)	57.0 (32.2)	68.7 (27.3)	69.5 (21.1)	0.629
Naive Resting (% B cells)	42.6 (13.2)	47.6 (34.9)	40.3 (5.0)	39.4 (16.3)	0.343
IgM-negative Naïve (% B cells)	23.4 (18.7) ^a^	4.5 (10.2) ^abc^	23.9 (21.0) ^b^	26.5 (12.0) ^c^	<0.001 *
Memory (%B cells)	20.3 (17.4)	12.8 (26.9)	22.8 (21.9)	21.5 (16.7)	0.429
Non-switched (% B cells)	7.4 (7.8) ^a^	2.0 (1.9) ^ab^	3.2 (7.0)	6.8 (7.7) ^b^	<0.001 *
Switched (% B cells)	9.7 (8.3)	7.8 (23.9)	11.1 (14.2)	10.5 (6.6)	0.879
DN (%B cells)	1.5 (1.4) ^a^	5.7 (7.6) ^ab^	2.5 (4.4)	2.0 (2.1) ^b^	<0.001 *
CD38− DN (% B cells)	0.9 (1.0) ^ab^	2.4 (4.2) ^a^	2.1 (3.0) ^b^	1.0 (1.3)	0.008 *
CD38+ DN (% B cells)	0.3 (0.5) ^a^	1.3 (1.8) ^abc^	0.4 (0.3) ^b^	0.5 (0.6) ^c^	<0.001 *
ASCs (%B cells)	2.1 (1.7) ^a^	7.2 (7.6) ^ab^	2.1 (1.3) ^b^	3.7 (3.1)	<0.001 *
EPBs (% B cells)	1.3 (1.3) ^ab^	5.3 (5.3) ^ac^	1.6 (1.3) ^c^	2.5 (1.7) ^b^	<0.001 *
PBs (% B cells)	0.3 (0.3)	0.6 (0.5)	0.2 (0.2)	0.4 (0.4)	0.060
Plasma cells (% B cells)	0.1 (0.1)	0.1 (0.2)	0.1 (0.1)	0.1 (0.1)	0.331

Data are analyzed using a Kruskal–Wallis test and Dunn’s post hoc tests and presented as the median and interquartile range (IQR). Small letters represent statistically significant differences between the two groups. * *p* < 0.05. Abbreviations: DN, double-negative B cells; ASCs, antibody-secreting cells; EPBs, early plasmablasts; PBs, plasmablasts; SLE, systemic lupus erythematosus; RA, rheumatoid arthritis; SpA, spondyloarthritis.

**Table 3 diagnostics-15-01710-t003:** Pearson correlation of peripheral B subpopulations with clinical and laboratory parameters of systemic lupus erythematosus.

Parameter		B Cells (% Lymphocytes)	Naive (% B Cells)	Naive Resting (% B Cells)	IgM-negative Naïve (% B Cells)	Memory (% B Cells)	Non-Switched Memory (% B Cells)	Switched Memory (% B Cells)	DN (% B Cells)	CD38− DN (% B Cells)	CD38+ DN (% B Cells)	ASCs (% B Cells)	EPBs (% B Cells)	PBs (% B Cells)	Plasma Cells (% B Cells)
SLEDAI-2K, points (*n* = 14)															
	r	−0.360	−0.411	−0.300	−0.159	−0.115	−0.196	−0.102	0.426	0.213	0.744	0.794	0.744	0.489	0.760
	*p*	0.205	0.145	0.297	0.587	0.697	0.502	0.728	0.129	0.464	0.002 *	0.001 *	0.002 *	0.090	0.002 *
Anti-dsDNA, WHO unit/mL (*n* = 10)															
	r	−0.516	−0.149	0.108	−0.672	0.011	−0.077	0.038	0.472	0.400	0.154	−0.020	−0.030	−0.022	0.026
	*p*	0.127	0.682	0.766	0.033 *	0.976	0.832	0.917	0.168	0.252	0.672	0.957	0.935	0.952	0.943
Creatinine, µmol/L (*n* = 14)															
	r	−0.308	−0.066	−0.159	0.218	−0.004	−0.087	−0.003	0.117	0.027	0.344	0.047	0.041	0.504	−0.101
	*p*	0.284	0.821	0.586	0.453	0.989	0.767	0.991	0.690	0.927	0.228	0.873	0.890	0.066	0.732
eGFR, mL/min/1.73 m^2^ (*n* = 14)															
	r	0.423	−0.218	−0.227	−0.088	0.131	−0.159	0.207	0.205	0.170	0.036	0.155	0.122	−0.353	0.077
	*p*	0.132	0.454	0.434	0.764	0.656	0.588	0.477	0.483	0.561	0.902	0.596	0.678	0.215	0.793
C3, mg/dL (*n* = 14)															
	r	−0.260	−0.134	−0.166	0.297	−0.170	−0.022	−0.206	0.229	0.175	0.177	0.417	0.452	0.035	0.387
	*p*	0.369	0.648	0.571	0.302	0.561	0.941	0.479	0.431	0.549	0.545	0.137	0.104	0.905	0.171
C4, mg/dL (*n* = 14)															
	r	−0.348	−0.095	−0.191	0.217	−0.068	−0.166	−0.047	0.218	0.148	0.320	0.152	0.157	0.370	0.042
	*p*	0.223	0.747	0.514	0.457	0.817	0.570	0.873	0.455	0.614	0.265	0.603	0.593	0.192	0.886
SLE duration, years (*n* = 14)															
	r	0.305	0.076	−0.045	0.155	0.086	0.057	0.080	−0.137	0.034	−0.375	−0.221	−0.192	0.284	−0.472
	*p*	0.290	0.797	0.880	0.596	0.769	0.845	0.786	0.641	0.907	0.186	0.448	0.510	0.326	0.089
Spot UPCR, mg/g (*n* = 14)															
	r	−0.366	−0.221	−0.308	0.162	0.048	−0.152	0.071	0.202	0.149	0.350	0.244	0.227	0.734	−0.202
	*p*	0.199	0.448	0.284	0.579	0.871	0.605	0.810	0.488	0.611	0.220	0.400	0.436	0.003 *	0.489

* *p* < 0.05. Abbreviations: SLEDAI-2K, systemic lupus erythematosus disease activity index 2000; r, correlation coefficient; *p*, *p*-value; DN, double-negative B cells; ASCs, antibody-secreting cells; EPBs, early plasmablasts; PBs, plasmablasts; eGFR, estimated glomerular filtration rate; SLE, systemic lupus erythematosus; UPCR, urine protein/creatinine ratio.

**Table 4 diagnostics-15-01710-t004:** Sensitivity, specificity, PPV, NPV, and odds ratio of the scoring system of peripheral B cell subpopulations in all patients and patients with SLE, RA, and SPA.

	Sensitivity	Specificity	PPV	NPV	OR
All	70.4%	70.8%	66.7%	74.2%	5.75
SLE	92.9%	70.8%	40.6%	97.9%	31.47
RA	57.1%	70.8%	24.2%	100.0%	3.22
SPA	56.3%	70.8%	26.5%	97.9%	3.11

Abbreviations: PPV, positive predictive values; NPV, negative predictive values; OR, odds ratio; SLE, systemic lupus erythematosus; RA, rheumatoid arthritis; SPA, spondyloarthritis.

## Data Availability

Data are available from the corresponding author upon reasonable request.

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
