# Peer review of "Alternation in Peripheral B Cell Subpopulations Is a Potential Biomarker for Autoimmune Diseases—A Cross-Sectional Study"

_diagnostics, 2025, doi:10.3390/diagnostics15131710_

Round 1
Reviewer 1 Report
Comments and Suggestions for Authors
The paper of Ku et al. explores the potential use of peripheral B cell subpopulation profiles as biomarkers for autoimmune disorders. Through flow cytometry, the researchers examined blood samples from 54 individuals with autoimmune conditions and 65 healthy participants, uncovering notable differences in the distribution of certain B cell subsets, particularly double-negative (DN) B cells and antibody-secreting cells (ASCs). Additionally, the study presents a new scoring model based on changes in B cell subsets.
The scoring system is promising but exploratory and authors should explain more the rationale for using quartiles and how points were assigned to each variable.
Please clarify whether the patients included in the study were treatment-naïve. If not, it would be important to address in the discussion whether ongoing or prior therapies may have significantly influenced the distribution of B cell subpopulations, as this could represent a potential limitation of the study.
While this study provides valuable insights into peripheral B cell subpopulations in autoimmune diseases, it would be beneficial for the authors to briefly consider the potential impact of sex hormones and chromosomal differences on B cell function and autoimmunity. Given the well-established sex bias in autoimmune disease prevalence—where females are more frequently affected—emerging evidence points to the role of X-chromosome dosage and hormonal regulation in immune dysregulation. The authors are invited to include relevant references, including a 2020 Clinical and Experimental Immunology study reporting a higher prevalence of non-organ-specific autoantibodies in adult males with Klinefelter syndrome (47,XXY), which underscores how sex chromosome aneuploidies may contribute to increased autoimmune susceptibility.
Author Response
|
Response to Reviewer 1 Comments
|
||
|
1. Summary |
|
|
|
We greatly appreciate comments from the editors and reviewers and hope the revised manuscript could be accepted for publication. All coauthors have reviewed the manuscript and have contributed in a substantive and intellectual manner to the work described. Please find our detailed point-by-point responses below, with revisions highlighted in the re-submitted files using track changes.
|
||
|
2. Questions for General Evaluation |
Reviewer’s Evaluation |
Response and Revisions |
|
Does the introduction provide sufficient background and include all relevant references? |
Yes |
Please see the point-by-point response. |
|
Is the research design appropriate? |
Yes |
|
|
Are the methods adequately described? |
Can be improved |
|
|
Are the results clearly presented? |
Yes |
|
|
Are the conclusions supported by the results? |
Can be improved |
|
|
Are all figures and tables clear and well-presented? |
|
|
|
3. Point-by-point response to Comments and Suggestions for Authors |
||
|
Comments 1: The scoring system is promising but exploratory and authors should explain more the rationale for using quartiles and how points were assigned to each variable. |
||
|
Response 1: We thank the reviewer for this excellent suggestion. By dividing the data from the control group into four equal parts, we established key thresholds of each variable to differentiate the core data from outliers or extreme values. Points were then assigned based on the relative position of the data. A total of 16 variables were scored, with higher points indicating a more significant impact on B cell alterations. We have revised the Method section (Page 4, line 155-162)
|
||
|
Comments 2: Please clarify whether the patients included in the study were treatment-naïve. If not, it would be important to address in the discussion whether ongoing or prior therapies may have significantly influenced the distribution of B cell subpopulations, as this could represent a potential limitation of the study. |
||
|
Response 2: We appreciate the reviewer’s insightful comment. The patients included in this study were not treatment-naïve. Nine patients with rheumatoid arthritis (RA) and all 14 patients with systemic lupus erythematosus (SLE) were on low-dose oral corticosteroids (equivalent dose less than 0.3 mg/kg/day of prednisolone), while five RA patients were treated with methotrexate. Patients with spondyloarthritis (SpA) or axial spondyloarthritis (AS) primarily received NSAIDs. Three of the 16 SpA patients were also treated with secukinumab, and four received sulfasalazine. Among the six AS patients, four were treated with sulfasalazine as well. There is no evidence that low-dose oral corticosteroids, methotrexate, or sulfasalazine significantly alter the composition of B cell subsets. Therefore, their use is less likely to substantially alter B cell subset distributions and impact data interpretation. In contrast, secukinumab has been reported to increase the proportions of naïve and class-switched B cells while decreasing memory and non-switched B cells in patients with AS, resulting in a B cell profile that more closely resembles that of healthy individuals [1]. Although secukinumab may normalize B cell subsets, it was administered to only a small proportion of SpA patients (3 out of 16) in our study. Therefore, we believe that its impact on B cell subset distribution is minimal. The impact of treatment on our data has been addressed. We have revised the Method section (Page 5, lines 171-177) and Discussion section (Page 11, lines 339-345).
|
||
|
Comments 3: While this study provides valuable insights into peripheral B cell subpopulations in autoimmune diseases, it would be beneficial for the authors to briefly consider the potential impact of sex hormones and chromosomal differences on B cell function and autoimmunity. Given the well-established sex bias in autoimmune disease prevalence—where females are more frequently affected—emerging evidence points to the role of X-chromosome dosage and hormonal regulation in immune dysregulation. The authors are invited to include relevant references, including a 2020 Clinical and Experimental Immunology study reporting a higher prevalence of non-organ-specific autoantibodies in adult males with Klinefelter syndrome (47,XXY), which underscores how sex chromosome aneuploidies may contribute to increased autoimmune susceptibility. |
||
|
Response 3: We appreciate the reviewer’s comment. In response, we conducted a comparison of B cell subpopulations between males and females and found no significant sex-related differences in either the control or patient groups (all p > 0.05). While we acknowledge that sex hormones and chromosomal variations, such as those observed in Klinefelter syndrome (47,XXY), can influence immune responses and potentially contribute to autoimmune susceptibility, a brief mention of this in the Discussion section of our manuscript has been included. We have revised the Result section (Page 5, lines 177-178) and Discussion section (Page 10, lines 345-348). |
||
|
|
||
References
- Jiang, Y.; Yang, M.; Zhang, Y.; Huang, Y.; Wu, J.; Xie, Y.; Wei, Q.; Liao, Z.; Gu, J. Dynamics of Adaptive Immune Cell and NK Cell Subsets in Patients With Ankylosing Spondylitis After IL-17A Inhibition by Secukinumab. Front Pharmacol 2021, 12, 738316, doi:10.3389/fphar.2021.738316.
Reviewer 2 Report
Comments and Suggestions for Authors
The manuscript focuses on B cell alterations in autoimmune diseases. In the abstract, the authors should first emphasize the differences in pathogenesis among various autoimmune diseases before extensively discussing the potential role of B cell subsets.
In the methods section, the description of B cell subset identification is insufficient. The authors should clearly specify which surface markers were used for subset identification (e.g., CD27, CD21, or others). This information is essential for reproducibility and clarity.
In the results, the authors mention double-negative (DN) B cells. However, data on other B cell subsets in the analyzed patient groups are missing and should be included. Furthermore, the manuscript references correlations between peripheral B cells in SLE patients and disease activity, but it is unclear which subsets were analyzed. The authors should specify how disease activity was measured — was it SLEDAI-2K? Renal SLEDAI? Anti-dsDNA titers? Complement levels (C3, C4)?
The introduction is concise but adequately introduces the topic.
In the methods section, exclusion criteria need to be described more comprehensively. Were patients excluded due to pregnancy, the postpartum period, malignancy, recent infections, or the use of immunosuppressive therapy within the last 6 months?
In the discussion, the authors should compare their findings on non-switched memory (NSM) B cells - especially in SLE, with existing literature, such as the study: DOI: 10.5603/FHC.a2016.0005. Did the authors observe similarly diminished levels of NSM B cells in SLE patients?
In the results, compare the distribution of B cell subsets with disease activity parameters, as done in the study: doi:10.3390/medicina60121994. These comparisons should also be discussed in the discussion section.
Additionally, clarify whether there were differences in B cell subsets between SLE patients with and without lupus nephritis.
Finally, include a limitations section to acknowledge the study’s constraints, such as sample size, cross-sectional design, or potential confounders.
Reviewer 3 Report
Comments and Suggestions for Authors
I appreciated having the opportunity to review the paper about the diagnostic value of peripheral B cell subpopulations in patients with autoimmune diseases using flow cytometry and a novel scoring system, I just a have some suggestions that I think can help improve the manuscript overall.
-The authors should include additional technical details regarding flow cytometry, such as: what were the compensation controls used for multicolor analysis? Where the steps for antibody panel specificity validated? What would you consider the reproducibility measures?
Please clarify if normality tests were performed before using non-parametric tests.
- You briefly mention that patients were under treatment. As a clinician, I do think this aspect has to be bore clearer as, as you know, treatment might affected B cell subpopulation distributions, so it has to be more detailed.
Round 2
Reviewer 1 Report
Comments and Suggestions for Authors
The authors have answered all my questions
Reviewer 2 Report
Comments and Suggestions for Authors
I accept changes introduced by authors. Congratulations.